# Protective Effect of Minimally Invasive Approach on Postoperative Peak Transaminase Following Liver Resection: A Single-Center Propensity Score-Based Analysis

**DOI:** 10.3390/cancers16142605

**Published:** 2024-07-21

**Authors:** Francesco Ardito, Sara Ingallinella, Quirino Lai, Francesco Razionale, Davide De Sio, Caterina Mele, Simone Vani, Maria Vellone, Felice Giuliante

**Affiliations:** 1Hepatobiliary Surgery Unit, Foundation Policlinico Universitario A. Gemelli, IRCCS, 00168 Rome, Italy; ingallinella.sara@hsr.it (S.I.); francesco.razionale@guest.policlinicogemelli.it (F.R.); davide.desio@guest.policlinicogemelli.it (D.D.S.); caterina.mele@policlinicogemelli.it (C.M.); simone.vani@policlinicogemelli.it (S.V.); maria.vellone@unicatt.it (M.V.); felice.giuliante@unicatt.it (F.G.); 2Department of Translational Medicine and Surgery, Università Cattolica del Sacro Cuore, 00168 Rome, Italy; 3General Surgery and Organ Transplantation Unit, Sapienza University of Rome, Policlinico Umberto I, 00161 Rome, Italy; lai.quirino@libero.it

**Keywords:** minimally invasive liver resection, open liver resection, hepatic damage, postoperative ALT level, hepatic pedicle clamping, liver manipulation

## Abstract

**Simple Summary:**

The degree of liver damage after liver resection is generally assessed by serum ALT levels. Postoperative ALT levels may have a multifactorial cause correlated with the extent of resection, duration of surgery and of vascular clamping. Extensive and prolonged manipulation of the liver during open hepatectomy could also be correlated with hepatocyte injury. The aim of our study was to assess if a minimally invasive approach for liver resection, with less manipulation of the liver, may be associated with less transient hepatic damage and with consequent lower postoperative ALT levels than those detected after open hepatectomy. The results showed that liver resections performed using a minimally invasive approach were associated with significantly lower postoperative ALT values when compared with those performed by open approach. Moreover, the duration of hepatic pedicle clamping and multiple liver resections were independent predictors for high postoperative peak ALT levels on POD 1 and the minimally invasive approach showed a protective effect.

**Abstract:**

Background: Postoperative serum ALT levels are one of the most frequently used marker to detect liver tissue damage following liver resection. The aim of this study was to evaluate if minimally invasive liver surgery (MILS) may result in less hepatic injury than open hepatectomy by assessing the differences of postoperative ALT levels. Methods: Patients who underwent MILS between 2009 and 2019 at our unit were included and compared with open liver resections. Median ALT levels was measured on postoperative day (POD) 1, 3 and 5. Postoperative peak transaminase (PPT) of ALT was determined on POD 1. The stabilized inverse probability treatment weighing (SIPTW) process was used to balance the two groups. A multivariable logistic regression analysis was used to analyze factors associated with high PPT. Results: After SIPTW, 292 MILS were compared with 159 open resections. Median ALT levels on POD 1, 3 and 5 were significantly higher in the open group than in the MILS group (301 vs. 187, *p* = 0.002; 180 vs. 121, *p* < 0.0001; 104 vs. 60, *p* < 0.0001; respectively). At the multivariable logistic regression analysis, MILS showed a protective effect for high PPT. Conclusions: MILS was associated with significantly lower postoperative ALT levels compared with open liver resections. MILS showed a protective effect for high PPT.

## 1. Introduction

The role of liver surgery in the treatment of malignant and benign tumors has been well established [1,2]. Hepatic pedicle clamping is widely used to minimize blood loss during hepatectomy. Its safety and efficacy in providing a clear operative field and in facilitating parenchymal transection have been clearly demonstrated in the literature [3,4,5,6]. After liver resection, a transient increase of serum alanine aminotransferase (ALT) levels is generally observed in relation to the hepatic ischemia/reperfusion injury associated with the use of hepatic pedicle clamping [7,8,9]. Moreover, hepatocellular injury could be related to liver mobilization and to surgical compression maneuvers. For these reasons, the evaluation of postoperative serum ALT levels following elective liver resections is usually performed in clinical practice in order to monitor the hepatocellular damage [10]. Indeed, the observed increase in serum ALT levels is due to the cellular release of cytoplasmic enzymes associated with hepatic cell damage. Serum ALT levels on postoperative day 1 may be a sensitive marker of hepatocyte injury and they could be considered as a surrogate which reflects the hepatocellular damage associated with these operative variables [11,12].

The use of a minimally invasive approach (laparoscopic and robotic) in the field of liver surgery has increased dramatically in the past decade [13,14]. The advantages related to the minimally invasive approach, including decreased postoperative pain, shorter hospital stay, earlier return to previous activity and decrease of postoperative complications and intraoperative blood loss, have been well documented [15]. The less invasiveness associated with the laparoscopic or robotic approach to liver resections, including minimal abdominal incision and less manipulation and compression during liver mobilization’s maneuvers, may contribute to reducing the hepatic damage. This type of advantage has not been clearly defined in the literature.

The aim of this study was to evaluate if minimally invasive liver surgery (MILS) may result in less hepatic injury than open hepatectomy by assessing differences of postoperative serum ALT levels.

## 2. Materials and Methods

### 2.1. Study Design

Between January 2009 and January 2019, data from liver resections performed with a minimally invasive approach at our unit were collected in a prospective database and reviewed retrospectively. These data were compared with all consecutive open liver resections (OLR) performed during the last two years of the study (2018–2019) in the same unit.

The study population included 659 adult patients undergoing elective liver surgery. Patients undergoing liver resections for both benign and malignant indications were included in the analysis. Exclusion criteria were (a) diagnosis of perihilar cholangiocarcinoma, (b) associated vascular resection during surgery and (c) converted minimally invasive resection cases.

The endpoint of the study was to evaluate if the surgical approach (MILS vs. OLR) may be correlated with the postoperative serum level of the alanine aminotransferase (ALT).

### 2.2. Variables

Explanatory variables were preoperative (including age, gender, preoperative chemotherapy, indication for surgery and type of underlying liver disease) and intraoperative variables (including type of surgical approach, extent and type of liver resection, duration of surgery, requirement of pedicle clamping and duration of pedicle clamping, blood loss and associated red blood cells transfusions). Laboratory values, including ALT serum levels, were recorded for all patients before surgery, on postoperative day (POD) 1, 3 and 5.

### 2.3. Technical Approach

#### 2.3.1. Open Liver Resections (OLR)

Hepatectomies were classified according to the International Hepato-Pancreato-Biliary Association (IHPBA) terminology [16]. Major hepatectomies included resections of three or more segments. Our surgical technique for liver resection has been previously described [17]. Parenchymal transection was performed by the Cavitron ultrasonic surgical aspirator (CUSA 200; Valleylab, Boulder, CO, USA) and wet bipolar forceps. Hepatic pedicle clamping was not routinely started at the beginning of liver resection, but it was used when bleeding did not allow a clear view of the operative field. Pedicle clamping was intermittently performed, in alternating periods of 15 min of ischemia and 5 min of reperfusion. Vessels were sealed using bipolar forceps, clips or staplers, depending on their size. Complete mobilization of the right hemiliver was always performed for tumors located in the postero-superior segments (segment 7 and 8) as for right hepatectomies or for right posterior sectionectomies.

#### 2.3.2. Minimally Invasive Liver Surgery (MILS)

Our surgical technique for MILS has been previously described [18]. The patient was placed in the supine position with the first surgeon standing between the patient’s legs and two assistants on the left side [18]. Pedicle clamping was used if required, as during OLR. Hepatic transection was performed with the use of an ultrasonic dissector (Misonix SONASTAR^®^, Biomedica Italia S.r.l., Assago, Milano). Vessels were sealed using Caiman vessel sealer (Aesculap^®^; B. Braun, Tuttlingen, Germany) bipolar forceps, clips or staplers, depending on their size. Complete mobilization of the right hemiliver was always performed for the same indications of OLR.

### 2.4. Statistical Analysis

Stabilized inverse probability treatment weighting (SIPTW) was performed within each database to balance patient characteristics between the two surgical groups (MILS and OLR). Propensity scores (PSs) were used to obtain estimates of the average treatment effect using a logistic model with the two cohorts of patients. Preoperative and intraoperative variables were included in the PS model.

After performing the PS calculation, each patient was weighted by the inverse of the probability of their treatment option (weight = 1/propensity score). The weights were stabilized by multiplying the original weights with a constant, which was equal to the expected value of being in the MILS or OLR cohort, respectively. After SIPTW, the baseline characteristics were balanced in each of the two databases, and patients were pooled for further analysis.

Continuous variables are expressed as a median with interquartile range (IQR). Discrete variables are expressed as counts (n) and percentages (%). Dependent variables were the study endpoints.

According to the results of the paper by Boleslawski E et al. [10], the cut-off of postoperative peak transaminase (PPT) of ALT on POD 1 was 336 IU/L. A multivariable logistic regression analysis was used to identify independent risk factors for high values of PPT-ALT on POD 1 (values > 336 IU/L). A *p*-value < 0.05 was considered statistically significant.

Postoperative results included 90-day mortality, complications, postoperative stay and need for re-intervention. Complications were scored according to the Clavien–Dindo grading system [19]. Major complications were defined as Clavien–Dindo grade ≥ 3.

## 3. Results

### 3.1. Patient’s Characteristics of the Entire Population

Between January 2009 and January 2019, 659 patients were scheduled to undergo liver resection. Of these, 606 patients fulfilled the inclusion criteria (Figure 1): 348 patients underwent OLR, while 258 underwent MILS (Table 1).

In the entire analyzed population, 100 patients (16.5%) were resected for benign disease. In the MILS group, the rate of resected patients for benign disease was significantly higher than that in the OLR group (26.4% vs. 9.2%, respectively; *p* < 0.001). In patients where malignant disease was the indication for surgery, metastases were the first indication for OLR, significantly more frequent than in the MILS group (73.0% vs. 45.3%, respectively; *p* < 0.001). HCC was the first indication in the MILS group, significantly more frequent than in the OLR group (24.4% vs. 10.6%, respectively; *p* < 0.001). Rates of major liver resections and of multiple liver resections were significantly more frequent in the OLR group than in the MILS group (30.7% vs. 9.3% and 23.6% vs. 3.1%, respectively; *p* < 0.001).

The use of pedicle clamping was significantly more frequent in the OLR group than in the MILS group (84.5% vs. 76.0%, respectively; *p* = 0.008). Rate of intraoperative blood transfusions in the OLR group was significantly higher than that in the MILS group (7.5% vs. 0.4%, respectively; *p* < 0.0001). Median duration of surgery was significantly higher in the OLR group than in the MILS group (390 min vs. 300 min; *p* < 0.0001).

### 3.2. Patient’s Characteristics after the Stabilized Inverse Probability Treatment Weighting (SIPTW) Process

SIPTW was performed to balance patients’ characteristics between the two cohorts. A total of 451 patients were included in the study after SIPTW was applied: 159 patients were included in the OLR group and 292 patients were included in the MILS group (Table 2).

Demographics, indications for surgery and intraoperative characteristics were now balanced between the two analyzed cohorts.

### 3.3. ALT Serum Levels on POD 1, 3 and 5

Variations of median ALT serum levels during the postoperative course according to the surgical approach are shown in Figure 2.

Patients undergoing MILS presented significantly lower ALT serum levels on POD 1, 3 and 5 if compared with OLR (Table 3a). These results were confirmed after the SIPTW process (Table 3b).

Seventy patients (11.5% of the entire population) underwent liver resection with pedicle clamping time ≥ 120 min. Of these, 45 patients (12.9%) underwent OLR and 25 patients (9.7%) underwent MILS. The rate of clamping time ≥ 120 min was not significantly different between the MILS and OLR groups (*p* = 0.21) (Table 1). Postoperative ALT serum levels were significantly lower on POD 1, 3 and 5 after MILS than after OLR (Table 4a). These differences were also confirmed after the SIPTW process but without statistical significance for the values detected on POD 1 and on POD 3 (Table 4b).

### 3.4. Postoperative Peak Transaminase (PPT)

Multivariable logistic regression analysis was performed to find independent predictors for high postoperative peak transaminase (PPT) of ALT on POD 1 (values > 336 IU/L). The results of this analysis are reported in Table 5. The minimally invasive approach showed a protective effect for high PPT-ALT on POD 1. Age, duration of pedicle clamping and the number of liver resections are independent risk factors for high PPT-ALT on POD 1.

### 3.5. Postoperative Results

In the series after SIPTW, the 90-day postoperative mortality was 0.4% (two patients), not significantly different between the OLR group and the MILS group (1.3% vs. 0; respectively; *p* = 0.12). Major complications (Clavien–Dindo grade ≥ 3) were observed in 34 patients (7.5%). The rate of major complications was significantly higher in the OLR group than that observed in the MILS group (15.7% vs. 3.1%, respectively; *p* < 0.001). The median postoperative hospital stay was significantly longer in the OLR group than that observed in the MILS group (8 days [7,8,9,10] vs. 5 days [4,5,6], respectively; *p* < 0.001). Fifteen patients (3.3%) underwent re-intervention. This rate was not significantly different between the two groups (5.1% in the OLR group vs. 2.4 in the MILS group; *p* = 0.17).

In order to evaluate a correlation between postoperative peak transaminase (PPT) of ALT on POD 1 and the postoperative results, a multivariable logistic regression analysis was performed (Table 6). High PPT-ALT on POD1 was an independent predictor of major complications and of re-interventions.

Variables were initially included in all the models and then removed using a backward Wald method: age, male sex, MILS, high PPT-ALT on POD 1 (ALT > 336 IU/L), benign disease, cholangiocarcinoma, HCC, metastases, major resection, multiple resections, duration of surgery, blood transfusions, pedicle clamping and clamping time ≥ 120 min.

## 4. Discussion

Our study showed that liver resections performed using a minimally invasive approach were associated with significantly lower median postoperative ALT values when compared with those performed with open approach.

Postoperative serum ALT levels are one of the most frequently used surrogate endpoints in several liver surgery studies [10,11]. Specifically, the postoperative peak transaminase (PPT) of ALT is the most frequently used marker to detect liver tissue damage [9,20].

Prospective studies have demonstrated that vascular clamping techniques, in particular hepatic pedicle clamping, are effective and safe in limiting bleeding during parenchymal transection [3,4,5,6]. The number of MILS to treat malignant and benign disease has significantly increased in the past decade [13,14]. Hepatic pedicle clamping is more commonly used during MILS for its effectiveness in reducing intraoperative blood loss and in providing a clear operative field during parenchymal transection [21].

On the other hand, hepatic pedicle clamping has been considered to have a great impact on the transient postoperative peak transaminase of ALT because it causes an ischemic hepatic damage that may be aggravated by the ischemia–reperfusion liver injury. In the review by Guo et al. [8], the authors evaluated all published data on postoperative peak ALT values after liver resection with hepatic pedicle clamping. The authors showed that the use of vascular clamping during hepatectomy was associated with significantly elevated postoperative peak ALT levels [8].

However, postoperative peak ALT levels might have a multifactorial cause strictly correlated with the extent of resection, the duration of surgery and the duration of ischemia reperfusion injury induced by vascular inflow occlusion [12,22]. In our retrospective study, these operative factors were not balanced between open surgery and minimally invasive surgery, where the rate of major liver resections was significantly higher in the OLR group than that in the MILS group. Moreover, the rate of pedicle clamping and its duration were significantly higher in the OLR group than in the MILS group. Finally, the duration of surgery was significantly higher in the OLR group than in the MILS group. For these reasons, the stabilized inverse probability treatment weighting (SIPTW) process was useful in order to balance patient characteristics between the two treatment strategies (MILS and ORL). After this process, demographics, indications for surgery and intraoperative characteristics were balanced between the two analyzed cohorts. In this study, we excluded patients who were converted from MILS to OLR in order to evaluate the real impact of pure MILS on the postoperative serum ALT levels.

The results of our study showed that in a large cohort of patients (606 patients), the median serum ALT levels detected on postoperative days 1, 3 and 5 were significantly lower following MILS than those following OLR. These results were found to be statistically significant both in the entire population and after the SIPTW process where the intraoperative factors (extent of resection, duration of surgery and duration of vascular inflow occlusion) were balanced.

Additionally, in a previous study by Doi et al. [20], the authors showed that postoperative ALT levels were significantly lower in the laparoscopic group than in the open group. However, this was a specific analysis focused only on few patients, all resected with hepatic pedicle clamping, with a cumulative Pringle time ≥ 120 min [20]. In our study, we evaluated this specific issue by assessing the effect of the duration of hepatic pedicle clamping on the postoperative ALT serum values. Our results confirmed that liver resections with a hepatic pedicle clamping time ≥ 120 min, performed with a minimally invasive approach, were associated with significantly lower ALT levels on POD 1 than those performed with an open approach.

These results may be related to other intraoperative factors, different from hepatic pedicle clamping, extent of resection and duration of surgery. Interestingly, it has been documented that serum markers of liver injury, such as transaminases levels, may increase before liver transection and before hepatic pedicle clamping, suggesting that factors other than ischemia–reperfusion may cause hepatic damage during liver surgery [23]. Indeed, extensive and prolonged manipulation of the liver during open hepatectomy might be correlated with hepatocyte injury, causing a transient postoperative increase of serum ALT levels. Historical studies have shown that simply retracting the liver with a valve during hiatal hernia repair was associated with postoperative cytolysis [23]. Previous studies showed that liver mobilization using an open approach with compression and manipulation of the liver may immediately cause hepatocellular damage and liver inflammation with a consequent increase of serum ALT levels [23,24,25]. Although the underlying pathophysiological mechanism remains unclear, liver manipulation is demonstrated as a leading cause of hepatocyte injury during open liver surgery [23,24,25].

The aim of our study was to assess if the minimally invasive approach for liver resection that requires less manipulation of the liver may be associated with less transient hepatic damage that is documented by significantly lower postoperative ALT levels than those detected after open hepatectomy.

According to the paper by Boleslawski E et al. [10], in our analysis we set the cut-off of postoperative peak transaminase (PPT) of ALT on POD 1 at 336 IU/L. A multivariable logistic regression analysis was used to identify independent risk factors for high values of PPT-ALT on POD 1 (values >336 IU/L). The results of this analysis confirmed that the duration of hepatic pedicle clamping and multiple liver resections were independent predictors for high postoperative peak transaminase (PPT) of ALT on POD 1. On the other hand, the interesting result was that the minimally invasive approach showed a protective effect for high PPT-ALT on POD 1 (OR = 0.399; *p* < 0.001).

Some papers have investigated if different surgical techniques of parenchyma transection may have an impact on intraoperative and postoperative results, and specifically on the postoperative peak transaminase levels [26,27,28,29]. In our technique, parenchyma transection is always performed by the ultrasonic dissector both in OLR and in MILS with a good level of performance and effectiveness [30]. A prospective randomized study by Lesurtel et al. [26] compared four different transection techniques in open liver resection (the clamp crushing technique with Pringle maneuver versus CUSA versus Hydrojet versus dissecting sealer without Pringle maneuver). The results of this study showed that the postoperative peak of transaminase AST and ALT was not significantly different among the four groups. These results were confirmed in a recent study focusing on transection techniques in pediatric major hepatectomy [29]. In this study, three liver transection techniques were compared (CUSA, LigaSure™ and stapler hepatectomy). In addition, in this study, peak AST and ALT values were not significantly different between the groups.

Finally, several studies have investigated the relevance of postoperative ALT levels on morbidity after liver resection, with controversial results [10,22,31]. This issue was not our end-point. However, it should be highlighted that in our analysis, a high PPT-ALT on POD 1 was an independent predictor of major complications and of re-interventions.

Our results may suggest that postoperative serum peak ALT levels could be significantly different following liver resection according to the type of approach (minimally invasive vs. open). This type of bias, due to the constant increase of the rate of MILS in all hepatobiliary centers, should be considered in future trials on liver surgery focusing on the evaluation of hepatic damage and of postoperative serum transaminases levels.

This study has several limitations. First, our study was a retrospective single-center study where the two groups of liver resections (MILS vs. OLR) from the initial cohort of patients were not balanced according to several intraoperative factors. Further assessment, including randomized prospective trials, may be required to confirm our results. Second, this series included patients resected between 2009 (at the beginning of our learning curve of minimally invasive approach) and 2019. During this period of time, indications for MILS and operative results following MILS changed and evolved.

## 5. Conclusions

This study showed that patients undergoing MILS presented significantly lower serum ALT levels during the postoperative course if compared with the OLR group after balancing the two groups according to the extent of resection, duration of surgery and duration of hepatic pedicle clamping. Moreover, the use of a minimally invasive approach was associated with a protective effect for a high postoperative peak of ALT levels.

## Figures and Tables

**Figure 1 cancers-16-02605-f001:**
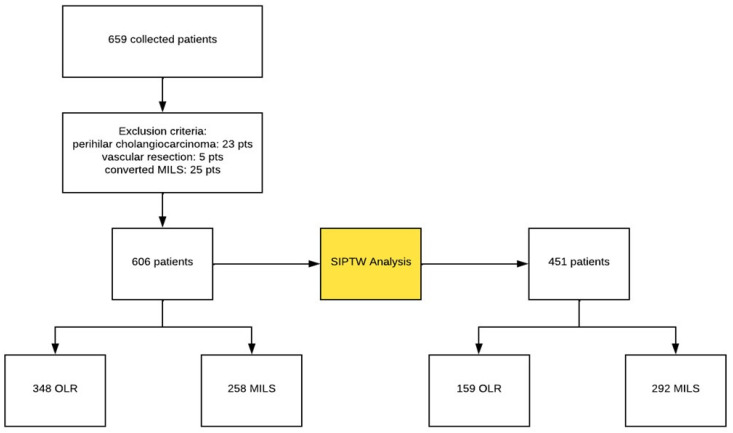
The patient’s selection process. SIPTW: stabilized inverse probability treatment weighting; MILS: minimally invasive liver surgery; OLR: open liver resections.

**Figure 2 cancers-16-02605-f002:**
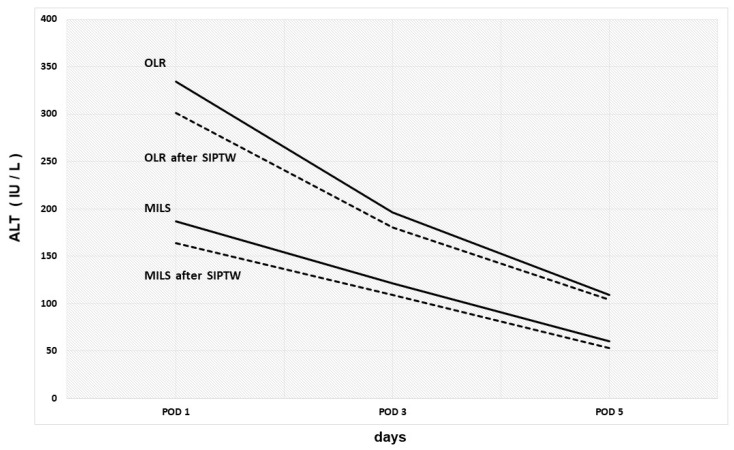
Variations of median ALT serum levels during the postoperative course according to the surgical approach before and after SIPTW process.

**Table 1 cancers-16-02605-t001:** Characteristics of the entire population (606 patients) according to the surgical approach (OLR vs. MILS).

	OLR (348)	MILS (258)	*p*-Value
Male, no. (%)	204 (58.6)	155 (60.1)	0.78
Female, no. (%)	144 (41.4)	103 (39.9)	
Age, median (IQR), yr	64 (56–70)	63 (53–71)	0.10
Benign disease, no. (%)	32 (9.2)	68 (26.4)	<0.001
Malignant disease, no. (%)	316 (90.8)	190 (73.6)	
Metastases	254 (73.0)	117 (45.3)	<0.001
HCC	37 (10.6)	63 (24.4)	<0.001
Cholangiocarcinoma	25 (7.2)	10 (3.9)	0.08
Major resection, no. (%)	107 (30.7)	24 (9.3)	<0.001
Multiple resections (no. ≥ 3), no. (%)	82 (23.6)	8 (3.1)	<0.001
Use of pedicle clamping, no. (%)	294 (84.5)	196 (76.0)	0.008
Clamping time, median (IQR), minutes	67 (45–97)	60 (31–96)	0.03
Clamping time ≥ 120 min, no. (%)	45 (12.9)	25 (9.7)	0.21
Intraoperative blood transfusions, no. (%)	26 (7.5)	1 (0.4)	<0.001
Duration of surgery, median (IQR), minutes	390 (305–502)	300 (220–400)	<0.001

OLR: open liver resections; MILS: minimally invasive liver surgery; IQR: interquartile range.

**Table 2 cancers-16-02605-t002:** Patient’s characteristics after the stabilized inverse probability treatment weighting (SIPTW) process.

	OLR (159)	MILS (292)	*p*-Value
Male, no. (%)	93 (58.5%)	165 (56.5%)	0.69
Female, no. (%)	66 (41.5%)	127 (43.5%)	
Age, median (IQR), yr	64 (55–70)	63 (51–72)	0.18
Benign disease, no. (%)	28 (17.6%)	64 (21.9%)	0.33
Malignant disease, no. (%)	130 (82.0%)	227 (77.7%)	
Metastases	97 (61.0%)	149 (51.0%)	0.06
HCC	25 (15.7%)	59 (20.2%)	0.31
Cholangiocarcinoma	9 (5.7%)	20 (6.9%)	0.69
Major resection, no. (%)	34 (21.4%)	49 (16.8%)	0.25
Multiple resections (no. ≥ 3), no. (%)	23 (14.6%)	29 (9.9%)	0.17
Use of pedicle clamping, no. (%)	132 (83.0%)	231 (79.1%)	0.38
Clamping time, median (IQR), minutes	58 (20–89)	47 (15–88)	0.39
Clamping time ≥ 120 min, no. (%)	19 (11.9)	41 (14.0)	0.57
Intraoperative blood transfusions, no. (%)	7 (4.4%)	14 (4.8%)	1.00
Duration of surgery, median (IQR), minutes	346 (287–477)	340 (244–480)	0.27

OLR: open liver resections; MILS: minimally invasive liver surgery; IQR: interquartile range.

**Table 3 cancers-16-02605-t003:** (**a**) Postoperative serum ALT levels in the entire population according to the surgical approach (OLR vs. MILS). (**b**) Postoperative serum ALT levels after SIPTW process according to the surgical approach (open vs. minimally invasive).

(a)			
	OLR (348)	MILS (258)	*p*-Value
**ALT levels, median (IQR)**			
POD 1	334 (207–570)	164 (94–324)	<0.0001
POD 3	196 (118–326)	109 (58–193)	<0.0001
POD 5	109 (64–55)	53 (34–01)	<0.0001
**(b)**			
	**OLR (159)**	**MILS (292)**	** *p* ** **-Value**
**ALT levels, median (IQR)**			
POD 1	301 (192–566)	187 (104–349)	0.002
POD 3	180 (114–308)	121 (65–213)	<0.0001
POD 5	104 (64–53)	60 (37–15)	<0.0001

OLR: open liver resections; MILS: minimally invasive liver surgery.

**Table 4 cancers-16-02605-t004:** (**a**) Postoperative serum ALT levels in the entire population following pedicle clamping time ≥ 120 min, according to the surgical approach (OLR vs. MILS). (**b**) Postoperative serum ALT levels after the SIPTW process following pedicle clamping time ≥ 120 min, according to the surgical approach (OLR vs. MILS).

(a)			
	OLR (n = 45)	MILS (n = 25)	*p*-Value
**ALT levels, median (IQR)**			
POD 1	586 (411–952)	346 (291–584)	0.004
POD 3	309 (218–515)	225 (162–357)	0.047
POD 5	154 (107–239)	121 (74–178)	0.08
**(b)**			
	**OLR (n = 19)**	**MILS (n = 41)**	***p*-Value**
POD 1	730 (436–1302)	611 (324–1251)	0.15
POD 3	373 (246–749)	351 (197–429)	0.07
POD 5	196 (132–278)	176 (114–233)	0.02

**Table 5 cancers-16-02605-t005:** Multivariable logistic regression analysis for predictors of high PPT-ALT on POD 1.

	Beta	SE	Wald	OR	Lower	Upper	*p*-Value
MILS	−0.920	0.244	14.223	0.399	0.247	0.643	<0.001
Duration of pedicle clamping	0.022	0.003	63.078	1.022	1.016	1.027	<0.001
Age	−0.025	0.009	6.693	0.976	0.958	0.994	0.010
Multiple resections (no. ≥ 3)	1.063	0.385	7.623	2.894	1.361	6.152	0.006

PPT: Postoperative peak transaminase; MILS: minimally invasive liver surgery.

**Table 6 cancers-16-02605-t006:** Multivariable logistic regression analysis for predictors of major complications, re-intervention and prolonged postoperative stay (>10 days).

	Beta	SE	Wald	OR	Lower	Upper	*p*-Value
Clavien–Dindo ≥ 3							
MILS	−1.72	0.57	9.23	0.18	0.06	0.54	0.002
High PPT-ALT on POD 1	0.99	0.51	3.84	2.69	1.01	7.24	0.049
Re-intervention							
High PPT-ALT on POD 1	2.88	0.98	8.66	17.81	2.62	121.31	0.003
Benign disease	2.23	0.87	6.60	9.32	1.70	51.10	0.01
Cholangiocellular cancer	2.62	1.18	4.98	13.77	1.38	137.89	0.03
Age, yr	0.07	0.03	4.59	1.08	1.01	1.15	0.03
Prolonged postoperative stay (>10 days)							
MILS	−2.99	0.78	14.87	0.05	0.01	0.23	<0.001
Duration of surgery	0.004	0.002	5.60	1.00	1.001	1.01	0.02

PPT: postoperative peak transaminase; MILS: minimally invasive liver surgery.

## Data Availability

The original contributions presented in the study are included in the article, further inquiries can be directed to the corresponding author.

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
