# Peer review of "Protective Effect of Minimally Invasive Approach on Postoperative Peak Transaminase Following Liver Resection: A Single-Center Propensity Score-Based Analysis"

_cancers, 2024, doi:10.3390/cancers16142605_

Round 1

Reviewer 1 Report

Comments and Suggestions for Authors

Dear authors

The paper is interesting and the methods are very well described. 

ALT is an easy marker of post-operative hypoxia - it is not the best and would improve your work if you have other factors.

SIPTW was crucial to have good conclusions. 

Although you did not perform that analysis, could you tell us:

The median days in the ward after each surgery and if there is a direct relationship with ALT? 

What is the mortality rate? What is the rate of technical success?

And the need for re-intervention?

Author Response

We thank the Reviewer for his detailed analyses of our manuscript and for his constructive criticisms and advices. We have modified the manuscript according to his comments and questions.

All the new inclusions and changes performed in the text following the comments are highlighted in yellow.

REPLY TO REVIEWER #1:

Comments and Suggestions for Authors

Dear authors

The paper is interesting and the methods are very well described.

ALT is an easy marker of post-operative hypoxia - it is not the best and would improve your work if you have other factors.

SIPTW was crucial to have good conclusions.

Although you did not perform that analysis, could you tell us:

The median days in the ward after each surgery and if there is a direct relationship with ALT?

What is the mortality rate? What is the rate of technical success?

And the need for re-intervention?

Response: This is a very interesting point. Results in the literature are controversial as we discussed in the Discussion section. This was not our end-point in this study. In the Results section (lines 271-291) we included the postoperative results of our series and we performed a new multivariable regression analysis in order to evaluate the impact of high postoperative peak of ALT on POD 1 on postoperative results. The results are interesting and they highlight the impact of postoperative peak of ALT on postoperative complications and the need for re-intervention. New details in the statistical analysis were also included (lines 149-151). A comment was included in the discussion section (lines 371-373).

Reviewer 2 Report

Comments and Suggestions for Authors

The authors present a very interesting study, which demonstrates the benefits of minimally invasive surery in liver resection for benign lesions, underlining the fact thata lower manipulation rate during laparoscopy is associated with lower postoperative values of transaminases. However, I would have one question for the authors: does the device used for parenchymal transection could also explain the differences? 

Meanwhile, I would suggest to increase the number of references (over 30)

Comments on the Quality of English Language

Minor English corrections are needed

Author Response

We thank the Reviewer for his comments. We revised the manuscript according to his questions.

All the new inclusions and changes performed in the text following the comments are highlighted in yellow.

REPLY TO REVIEWER #2:

Comments and Suggestions for Authors

The authors present a very interesting study, which demonstrates the benefits of minimally invasive surgery in liver resection for benign lesions, underlining the fact that lower manipulation rate during laparoscopy is associated with lower postoperative values of transaminases. However, I would have one question for the authors: does the device used for parenchymal transection could also explain the differences?

Meanwhile, I would suggest to increase the number of references (over 30)

Response: thank you for this interesting question. This issue has been investigated in a very interesting prospective trial by Lesurtel et al. In this study the authors showed no significant differences in peak transaminase levels according to the type of transection technique. We included a comment in the discussion section (lines 369-381).

The number of references has been increased.

Round 2

Reviewer 1 Report

Comments and Suggestions for Authors

Dear authors

After some corrections, the paper has improved.